# Gender Differences in Association between Air Pollution and Daily Mortality in the Capital of the Green Lungs of Poland–Population-Based Study with 2,953,000 Person-Years of Follow-Up

**DOI:** 10.3390/jcm9082351

**Published:** 2020-07-23

**Authors:** Łukasz Kuźma, Krzysztof Struniawski, Szymon Pogorzelski, Hanna Bachórzewska-Gajewska, Sławomir Dobrzycki

**Affiliations:** 1Department of Invasive Cardiology, Medical University of Bialystok, 15-276 Bialystok, Poland; kstruniawski@gmail.com (K.S.); szymonpogo@gmail.com (S.P.); hgajewska@op.pl (H.B.-G.); kki@umb.edu.pl (S.D.); 2Department of Clinical Medicine, Medical University of Bialystok, 15-276 Bialystok, Poland

**Keywords:** air pollution, mortality, cardiovascular mortality, gender difference, sulfur dioxide

## Abstract

(1) Introduction: air pollution is considered to be one of the main risk factors for public health. According to the European Environment Agency (EEA), air pollution contributes to the premature deaths of approximately 500,000 citizens of the European Union (EU), including almost 5000 inhabitants of Poland every year. (2) Purpose: to assess the gender differences in the impact of air pollution on the mortality in the population of the city of Bialystok—the capital of the Green Lungs of Poland. (3) Materials and Methods: based on the data from the Central Statistical Office, the number—and causes of death—of Białystok residents in the period 2008–2017 were analyzed. The study utilized the data recorded by the Provincial Inspectorate for Environmental Protection station and the Institute of Meteorology and Water Management during the analysis period. Time series regression with Poisson distribution was used in statistical analysis. (4) Results: A total of 34,005 deaths had been recorded, in which women accounted for 47.5%. The proportion of cardiovascular-related deaths was 48% (*n* = 16,370). An increase of SO_2_ concentration by 1-µg/m^3^ (relative risk (RR) 1.07, 95% confidence interval (CI) 1.02–1.12; *p* = 0.005) and a 10 °C decrease of temperature (RR 1.03, 95% CI 1.01–1.05; *p* = 0.005) were related to an increase in the number of daily deaths. No gender differences in the impact of air pollution on mortality were observed. In the analysis of the subgroup of cardiovascular deaths, the main pollutant that was found to have an effect on daily mortality was particulate matter with a diameter of 2.5 μm or less (PM2.5); the RR for 10-µg/m^3^ increase of PM2.5 was 1.07 (95% CI 1.02–1.12; *p* = 0.01), and this effect was noted only in the male population. (5) Conclusions: air quality and atmospheric conditions had an impact on the mortality of Bialystok residents. The main air pollutant that influenced the mortality rate was SO_2_, and there were no gender differences in the impact of this pollutant. In the male population, an increased exposure to PM2.5 concentration was associated with significantly higher cardiovascular mortality. These findings suggest that improving air quality, in particular, even with lower SO_2_ levels than currently allowed by the World Health Organization (WHO) guidelines, may benefit public health. Further studies on this topic are needed, but our results bring questions whether the recommendations concerning acceptable concentrations of air pollutants should be stricter, or is there a safe concentration of SO_2_ in the air at all.

## 1. Background

Air pollution is caused by a complex mixture of various chemical substances, mainly sulfur dioxide (SO_2_), nitrogen dioxide (NO_2_), carbon monoxide (CO), benzo(a)pyrenes, and particulate matter with a diameter of 2.5 μm or less (PM2.5) and 10 μm or less (PM10). Some of them are formed directly during the burning process, which is associated with industries or heating devices at homes, or car traffic, while others are produced by photochemical reactions occurring in the air under the influence of ultraviolet radiation. Intensification of these processes and the atmospheric conditions influence the mixture of air pollutants and their potential effects on public health and life [1]. 

The influence of air pollution on human health began to gain attention during the early second half of the 20th century, after the Great Smog of London. During this episode, around 12,000 deaths occurred as a result of circulatory and respiratory insufficiency [2], and since then, the scientists had undertaken intensive researches into the impact of air pollutants on health. Currently, air pollution is considered one of the main risk factors for public health, in particular the disorders of the pulmonary and cardiovascular system and related mortality. These are also addressed in a number of guidelines, particularly those concerning the treatment of patients with cardiovascular and pulmonary diseases. According to the European Environment Agency (EEA), air pollution contributes to the premature death of approximately 500,000 citizens of the European Union (EU), including almost 5000 inhabitants of Poland, every year [3,4,5,6,7,8]. There are three main mechanisms of short-term harmful impact of air pollution: activation of the sympathetic system, generalized inflammatory response, and direct harmful effects of air pollutants on the circulatory system [8,9,10].

A majority of prior studies concerning the influence of air pollution on mortality were carried out in cities and areas where the level of air pollution is usually high or very high [11,12]. Bialystok, which is the subject of the present study, is a city situated in Poland in Eastern Europe with a population of almost 300,000 inhabitants (53.1% women, 46.9% men). It is the capital of Podlaskie Voivodeship with 1.5 million citizens. At present, individual industrial plants are located in the city, and the predominant source of air pollution is low–emission ones, particularly vehicle traffic (45–79%) and heating furnaces used at homes (16–24%) [13]. The city is considered as the Polish symbol of a clean environment. It is the capital of the region referred to as the Green Lungs of Poland, which are the areas characterized by unique natural features and rich biodiversity. Due to these values, Bialystok has become the first Polish city to be included in the Global Organization “Healthy Cities Network” project [14].

As there is a lack of comprehensive studies devoted to the effects of low-level air pollution and weather conditions on mortality, we decided to investigate these relationships in the city of Bialystok. We carried out separate analyses for male and female groups and determined the total mortality rate and mortality due to cardiovascular disease (CVD) in particular.

## 2. Aim of the Study

To assess the gender differences in the impact of air pollution on the mortality in the population of the city of Bialystok—the capital of the Green Lungs of Poland.

## 3. Patients and Methods

### 3.1. Mortality Data

Data on mortality were collected from the Statistical Office in Olsztyn. These include information on all the deaths recorded in the city of Białystok in the years 2008–2017.

The records included sex and age of people who had died, and the causes of deaths were classified according to codes in the International Classification of Diseases—10th Revision (ICD-10) [15]. We also separately collected the data for CVD-related mortality (ICD-10 from I.00 to I.99) for men and women.

Mean daily and yearly statistic of deaths were determined for the whole population and both the male and female populations. The daily numbers of deaths were used for statistical analysis.

No mortality data were found to be missing for the analyzed period.

### 3.2. Air Pollution and Meteorological Data

The data on the concentrations of air pollutants obtained from two stations of the Voivodeship Inspectorate for Environmental Protection in Białystok-Waszyngtona Street 16 and Warszawska Street 75—for the years 2008–2017 were utilized in the analysis. The concentrations of SO_2_ (Environment AF22E, Waszyngtona Street 16, Bialystok, Poland), NO_2_ (Teledyne API T200, Waszyngtona Street 16, Bialystok, Poland), PM2.5 (MetOne BAM 1020, Waszyngtona Street 16, Bialystok, Poland), and PM10 (TEOM 1405F, Warszawska Street 75, Bialystok, Poland) were analyzed using different analyzers. Stations are located in the center of the city with a maximum distance to the borders of 6 km. 

Daily meteorological data were obtained from the Institute of Meteorology and Water Management for the analyzed period. We used 24 h data on average temperature, average relative humidity, and average atmospheric pressure obtained from the weather station located in the center of the city with a maximum distance to the borders around 6 km.

Mean daily and yearly concentrations were determined for all the physical parameters. The daily values were used for statistical analysis. Exceedance of the concentration of air pollution was determined based on the guidelines of the World Health Organization (WHO) on air quality published in 2005 [16]. The 24 h and annual mean concentrations recommended by the WHO for PM10 are 50 µg/m^3^ and 20 µg/m^3^ and for PM2.5 are 25 µg/m^3^ and 10 µg/m^3^, respectively, while for SO_2_, the accepted 24 h mean concentration is 20 µg/m^3^. 

During the study, we found that 4.8% of the observations were missing. Days with missing data were excluded from the analysis.

### 3.3. Statistical Analysis

The distribution of variables was evaluated using the Kolmogorov–Smirnov test, while the two-tailed T-test was used for comparative analysis. Continuous variables were expressed as mean values with standard deviation. When appropriate, medians with interquartile ranges were used. Non-normally distributed data were compared with the Mann–Whitney test. Spearman’s rank correlation test was applied for evaluating the relationships between the levels of air pollutants.

Time series regression was used to assess the short-term effect of particles, concentrations of gases, and temperature on mortality. We used multivariable Poisson distribution. To minimize the collinearity effect, each air pollutant was modelled individually. Separate models were created for temperature with humidity and atmospheric pressure as covariates. The daily number of deaths was defined as dependent variable. The independent variables were daily concentrations of NO_2_, SO_2_, PM2.5, PM10, and temperature. To control for the long-term trend and seasonal effects we used time stratified model (simple indicator variables). The time interval for daily data was elapsed calendar month, which resulted in creating 120 strata [17]. We adjusted our model for the day of the week and public holidays. As co-founders, we used absolute value of change in maximum temperature from day-to-day, humidity, and atmospheric pressure, which were environmental risk factors for acute coronary syndromes and mortality in prior studies. The results were presented as relative risks (RRs) for a 10-μg/m^3^ increase in the concentration of particular matter (PM2.5 and PM10), 1-μg/m^3^ increase in the concentration of gases (SO_2_ and NO_2_), 10 °C decrease in temperature, and confidence interval (CI) from the 5th to the 95th percentile. 

We created a model for all patients and two gender-specific. In the next stage, we compared head to head the impact of air pollution on women and men. The differences in the relative risk between males and females were calculated according to Altman method [18]. The results were presented as ratios of relative risks (RRRs) and CI from the 5th to the 95th percentile. 

The threshold of statistical significance for all tests was set at *p* < 0.05. All analyses were performed with MS Excel and SPSS IBM Software. The study protocol was approved by the ethics committee of the Medical University of Bialystok (R-1–002/18/2019).

## 4. Results

Data on 16,154 (47.5%) deaths in the female population (mean/day 4.42, SD = 2.14) with mean age of 76.05 (SD = 15.48) and 17,851 (52.5%) deaths in the male population (mean/day 4.89, SD = 2.28) with mean age of 68.33 (16.69) recorded in the period between 2008 and 2017 were qualified for further analysis (*p* < 0.001) (Table 1).

In the case of women, an increase in the number of deaths was observed every year (1478 in 2008 vs. 1846 in 2017), and a clear trend of increase in the mean age of the deceased was also noted: 74.10 (SD = 16.67) in 2008 vs. 78.08 (SD = 14.40) in 2017. Similar observations of the increased number of deaths and the increase in the age of the deceased were noted in the male population (1683, mean age 67.33 (SD = 16.83) in 2008 vs. 1942, mean age 69.59 (SD = 16.29) in 2017) (Appendix A in Appendix A). The death rate of the analyzed population was 1279 and 1023 per 100,000 inhabitants per year in the case of men and women, respectively.

The proportion of deaths due to cardiovascular reasons was 52.32% (*n* = 8451) with the mean age of the deceased being 81.29 (SD = 8.99) in the female population vs. 44.36% (*n* = 7919) with the mean age being 72.18 (SD = 12.12) in the male population (*p* < 0.001) (Table 1 and Table 2).

The main causes of deaths in the female population were chronic ischemic heart disease (*n* = 1683; 10.42%) and cerebral infarction (*N* = 1617; 10.01%), while in the case of men the main causes were malignant neoplasm of the bronchus and lung (*n* = 1555; 8.71%) and chronic ischemic heart disease (*N* = 1530; 8.57%). The most frequent oncological cause of deaths in both the female and male population was malignant neoplasm of the bronchus and lung (4.29% vs. 8.71%, *p* < 0.001) (Table 2).

In the analysis of air pollution, the mean concentrations of SO_2_, NO_2_, PM2.5, and PM10, and the mean levels of humidity, temperature, and atmospheric pressure over 10 consecutive years starting from 2008 were taken into account. The mean daily concentration of PM2.5 was 20.35 µg/m^3^ (SD = 15.05), PM10 was 24.53 µg/m^3^ (SD = 15.23), NO_2_ was 14.40 µg/m^3^ (SD = 6.45), and SO_2_ was 3.29 µg/m^3^ (SD = 3.10). During analysis, it was observed that the 1-day upper limit of concentration recommended by the WHO guidelines for SO_2_ was sporadically exceeded (*N* = 18, 0.50% of analyzed days), while the daily upper limit for PM2.5 was exceeded on 688 days (23.60% of analyzed days). The daily upper limit for PM10 (50 µg/m^3^) recommended by the WHO guidelines was observed to be exceeded on 172 days (5.29%) (Table 3). The annual WHO guideline values for PM2.5 and PM10 were exceeded every year (Appendix A in Appendix A).

A moderate positive correlation was found between concentration of particulate matter and gases—PM2.5 vs. NO_2_: *R* = 0.53, *p* < 0.001; PM2.5 vs. SO_2_: *R* = 0.45, *p* < 0.001; PM10 vs. SO_2_: *R* = 0.40, *p* < 0.001; and PM10 vs. NO_2_: *R* = 0.49, *p* < 0.001. In addition, a weak correlation was found between the concentrations of gases—SO_2_ and NO_2_ (*R* = 0.28, *p* < 0.001). Moreover, a moderate negative correlation was found between the values of temperature and concentration of particulate matter and gases—PM2.5 vs. temperature: *R* = −0.47, *p* < 0.001; PM10 vs. temperature: *R* = −0.32, *p* < 0.001; SO_2_ vs. temperature: *R* = −0.33, *p* < 0.001; and NO_2_ vs. temperature: *R* = −0.41, *p* < 0.001 (Appendix A in Appendix A).

In Table 4, the impact of air pollution on short-term mortality in studied population is shown. A 1 μg/m^3^ increase of SO_2_ concentration (RR 1.07, 95% CI 1.02–1.12; *p* = 0.005) and a 10 °C decrease of temperature (RR 1.03, 95% CI 1.01–1.05; *p* = 0.005) were related to an increase in the number of daily deaths.

The detailed analysis by gender is presented in Table 5 and Table 6. In the female group, excluding the seasonal impact, a 1 μg/m^3^ increase of SO_2_ concentration (RR 1.05, 95% CI 1.01–1.10; *p* = 0.009) and a 10 °C decrease of temperature (RR 1.04, 95% CI 1.02–1.07; *p* = 0.003) were related to an increase in the number of daily deaths. In the analysis of the subgroup of cardiovascular deaths, no association with air pollutants was observed, and a 10 °C decrease of temperature was associated with an increase in daily mortality (RR 1.08, 95% CI 1.04–1.13; *p* < 0.001) (Table 5). In the male group, an effect of SO_2_ on mortality was noted; the RR for 1-µg/m^3^ increase of SO_2_ was 1.10 (95% CI 1.04–1.17; *p* = 0.002). In the subgroup analysis of cardiovascular deaths, the main pollutant that was found to have an effect on daily mortality was PM2.5; the RR for 10-µg/m^3^ increase of PM2.5 was 1.07 (95% CI 1.02–1.12; *p* = 0.01). No effect of temperature on mortality was observed in men (Table 6).

In Table 7, the comparison of relative risk according to Altman method between males and females is shown. No significant differences were found between the groups. 

## 5. Discussion

Over the years, a steady increase in the number of deaths had been observed in Białystok; in 2008, there were 3161, whereas in 2017 there were 3788 deaths. The increase in the number of deaths and the increasing mean age of the deceased in the Białystok population are aligned with a similar trend at both the national and the European level. According to the data of the Central Statistical Office, in 2018, 414,000 Polish citizens had died, which was 3.5% more than the preceding year. This trend had been continuing from approximately 2012 [19,20,21].

The death rate of the analyzed population was 1.25-fold higher in the case of men than women. Similar gender differences are reported in the European and global studies. Statistical results show that men die more frequently, both at the European level (standard death rate, SDR, 1250 deaths per 100,000 inhabitants) as well as in Poland (SDR 1616 deaths per 100,000 inhabitants). In the EU, the mean number of deaths (SDR) in the case of women is 817 per 100,000 inhabitants, while it is 941 per 100,000 inhabitants in Poland.

Population aging poses numerous challenges to the economy and the sector of health care services. Moreover, it is estimated that in the near future the Polish society will be among the oldest in the EU [20,21]. Changes in the age structure of society will also lead to a change in the causes of deaths. At present, CVDs are the predominant cause of deaths in Poland, with almost every second death occurring due to them. The present study showed that the major causes of deaths were CVDs; precisely, 52.32% of women and 44.36% of men (*p* < 0.001) died due to CVDs. This percentage varies between regions and countries. In the EU, the mean death rate due to cardiovascular reasons is 38,4% and 33.1% for women and men, respectively, while for the neighboring regions, such as Latvia and Lithuania, this rate is around 57% (64–66% for women, 48–50% for men) [22]. 

The second most frequent cause of deaths in the EU are malignant neoplasms (25.4%). Over the years, a decrease in the number of deaths due to oncological reasons had been observed; however, the number of oncological deaths in the male population remained at an approximately 40% higher level than in the female population. This index is also higher than in developed countries, where it is estimated at approximately 20% [23]. The reason for this finding might be the difference in tumor incidence between the male and female population and the greater prevalence of risk factors in the group of men, in particular smoking tobacco products.

The city of Bialystok has the 10th largest population in Poland and is characterized by lower SDR when compared to other major polish cities [24]. This suggests that the city itself, or Podlaskie Voivodeship on the whole, has a positive impact on the life expectancy of its residents, thanks to the abundant green areas and low level of industrialization, which leads to lower environmental pollution, and in particular better air quality, as compared to other regions [10]. However, the higher average age of the population predisposes them to the harmful effects of air pollutants [10,25,26].

In our analysis, the effect of SO_2_ was more pronounced in the male population, but the differences between males and females were insignificant. The exact pathophysiologic mechanisms responsible for the SO_2_ impact on mortality are hard to pinpoint, but SO_2_ inhalation may activate the events of pulmonary inflammation, leading to systemic inflammation, which promotes thrombosis and endothelial dysfunction. The inhaled SO_2_ dysregulates the sensory receptors in the lungs, resulting in an imbalance of the autonomous nervous system, favoring sympathetic paths and potentially leading to changes in pulse rate, vascular stenosis, and an increase in arterial pressure. 

The results obtained in the present study seem to be in line with the current literature. A study conducted in Beijing estimated that the mortality rate caused by the increase in SO_2_ concentration was 27,854 deaths per annum [27]. In a study that conducted a detailed analysis of the influence of SO_2_ in six metropolitan cities of China, the short-term influence on mortality associated with the increase of SO_2_ by 10 μg/m^3^ was estimated to range from 0.44% to 1.14% [11]. Another study assessed the influence of particulates and SO_2_ during winter on mortality due to lung cancer and showed that it was statistically significant (ranging from 0.93% to 7.16% for an increase by 10 μg/m^3^) [12]. 

Many prior studies have demonstrated the aforementioned relationship between air pollution and mortality, which is linked to the increase of not only SO_2,_ but also other components, such as NO_2_ and particulate matter [11,12,28,29,30]. 

Analysis of the subgroups of deaths due to cardiovascular reasons in the male group demonstrated the influence of PM2.5. The effect of PM2.5 on the occurrence of acute coronary events has already been well-documented [31,32,33], which can be directly linked to elevated mortality rates. PM2.5 penetrates the lungs and enters the blood, and through mechanisms such as elevated inflammation, oxidative stress, hypercoagulability, and endothelial dysfunction, as well as an increase in blood pressure and positive chronotropic response, it leads to heart overload, which may result in myocardial infarction. In addition, flue gas particles from engines often have volatile organic compounds adhered to them, which lead to the derangement of the atherosclerotic plaque. The formation of reactive oxygen species induced by air pollution may disturb vasodilation through NO, and it was also demonstrated that oxidative stress is primarily caused by the surface compounds adhering to the particulate matter [10,28,34].

Data on the influence of PM2.5 on mortality have been recorded in numerous studies conducted around the world [35,36,37]. In the present study, the proportion of deaths due to cardiovascular reasons was estimated to be 52.32% in the female population and 44.36% in the male population. Despite the higher percentage of deaths due to CVDs in the female population, no influence of PM2.5 on deaths was observed in this group, which is likely due to the high contribution of sudden cardiac deaths observed in the male population (10.3% vs. 4.1% of deaths due to acute coronary event and sudden deaths). Some previous studies have demonstrated that the female population is more susceptible to the influence of particulate matter [38,39]; however, the literature is dominated by reports showing a greater impact of air pollution on male population or the lack of gender differences [40,41,42,43].

Different research outcomes may stem from different forms of activity carried out by both groups (women and men) in different areas of the world, but also from different compositions of air pollution in individual climate zones. Moreover, what is also important is the varying prevalence of risk factors for death in different populations and exposure to non-environmental sources of particulate matter, or sources linked to work or daily activity.

Moreover, the present study demonstrated the influence of temperature on total mortality, both in the female and male population, as well as on cardiac mortality in the female population. These results have been already established in the scientific literature. A study analyzing the influence of temperature changes between subsequent days on mortality, covering cities in similar climatic zones on both hemispheres, demonstrated a significant impact of temperature fluctuation on total mortality and cardiovascular mortality. Liang et al. demonstrated that considerable daily fluctuations in temperature are linked to an increased number of admissions to emergency departments in hospitals due to acute coronary events. This phenomenon is explained by the temperature stimulation of dermal receptors, which activates the sympathetic nervous system, leading to increased concentration of catecholamines in the blood, increased arterial pressure, elevated heart rate, increased myocardial oxygen demand, and increased platelet activation, which altogether increase the risk of the exacerbation of coronary heart disease and the occurrence of acute coronary events [44,45,46].

## 6. Study Limitations

A major limitation of the study concerning the analysis of the causes of deaths due to cardiovascular reasons is the use of the so-called garbage codes for determining this group. It should be noted that the majority of useless codes are assigned to the individuals above 65 years, who constituted the bulk of the discussed population. An additional limitation hindering comparative analysis is the lack of standardized death rates for specific populations of women and men in the city of Bialystok

In our opinion, the limitation of the study is also the underdeveloped air pollution monitoring system of the city and the lack of constant monitoring of some pollutants (e.g., benzene or lead). Air pollution in urban areas is characterized by high spatial fluctuations in the levels of pollutants, which could affect the results. 

## 7. Conclusions

Air quality and atmospheric conditions had an impact on the mortality of Bialystok residents.The main air pollutant that influenced the mortality rate was SO_2_, and there were no gender differences in the impact of this pollutant. In the male population, an increased exposure to PM2.5 concentration was associated with significantly higher cardiovascular mortality.Our findings suggest that improving air quality, in particular, even with lower SO_2_ levels than currently allowed by WHO guidelines, may benefit public health. Further studies on this topic are needed, but our results bring questions whether the recommendations concerning acceptable concentrations of air pollutants should be stricter, or is there a safe concentration of SO_2_ in the air at all. 

## Figures and Tables

**Table 1 jcm-09-02351-t001:** Mortality in Bialystok in the years 2008–2017.

	Total Mortality (Female)	Total Mortality (Male)	*p*	Cardiovascular Mortality (Female)	Cardiovascular Mortality (Male)	*p*
Total, *n*;	16,154	17,851		8451	7919	
Mean age (SD)	76.05 (15.48)	68.33 (16.69)	<0.001	81.29 (8.99)	72.18 (12.12)	<0.001
Daily mean (SD)	4.42 (2.14)	4.89 (2.28)	<0.001	2.31 (1.55)	2.17 (1.51)	<0.001
Daily minimum	0	0		0	0	
Daily 1st quartile	3	3		1	1	
Daily median	4	5		2	2	
Daily 3rd quartile	6	6		3	3	
Daily maximum	15	14		10	9	
Daily IQR	3	3		0	2	

Abbreviations: IQR, interquartile range; N/A, not applicable; SD, standard deviation.

**Table 2 jcm-09-02351-t002:** Comparison of the main causes of deaths in Bialystok in female and male population in the years 2008–2017.

	Female	Male	*p*
Total, *N*; %	16,154; 100	17,851; 100	*N*/A
Cardiovascular mortality, *N*; %	8451, 52.32	7919, 44.36	<0.001
Chronic ischemic heart disease, *N*; %	1683; 10.42	1530; 8.57	<0.001
Cerebral infarction, *N*; %	1617; 10.01	1110; 6.22	<0.001
Heart failure, *N*; %	1370; 8.48	776; 4.34	<0.001
Acute myocardial infarction, *N*; %	537; 3.32	756; 4.24	<0.001
Intracerebral hemorrhage, *N*; %	476; 2.95	577; 3.23	0.12
Other reason mortality, *N*; %	7703, 47.68	9932, 55.64	<0.001
Neoplasm of bronchus and lung, *N*; %	693; 4.29	1555; 8.71	<0.001
Pneumonia, unspecified, *N*; %	467; 2,89	477; 2.67	0.22
Malignant neoplasm of colon, *N*; %	345; 2.14	426; 2.39	0.12
Senility, *N*; %	817; 5.06	278; 1.55	<0.001
Malignant neoplasm of prostate, *N*; %	0; 0.00	474; 2.66	*N*/A
Malignant neoplasm of breast, *N*; %	551; 3.41	0; 0.00	*N*/A

Abbreviation: *N*/A, not applicable.

**Table 3 jcm-09-02351-t003:** Statistics for daily concentrations of air pollutants and weather conditions in the period of 2008–2017.

	NO_2_ µg/m^3^	SO_2_ µg/m^3^	PM2.5 µg/m^3^	PM10 µg/m^3^	Temp. °C	RH %	Atm. P. hPa
No. of observations, *N*, %	3620, 99.09	3597, 98.45	2938, 80.43	3248, 88.91	3653, 100	3653, 100	3653, 100
Daily mean (SD)	14.40 (6.45)	3.29 (3.10)	20.35 (15.06)	24.53 (15.23)	7.80 (8.70)	81.11 (12.0)	997.31(8.51)
Daily minimum	1.40	0.00	1.16	2.69	−23.90	38.30	959.60
Daily 1st quartile	9.88	1.43	10.77	14.80	1.50	73.60	992.10
Daily median	13.14	2.62	15.90	21.00	7.80	83.11	997.30
Daily 3rd quartile	17.77	4.21	24.79	30.00	14.80	90.60	1002.60
Daily maximum	70.05	37.59	139.22	192.50	26.60	100.00	1025.50
Daily IQR	7.89	2.78	14.02	15.20	13.30	17.00	10.50
Exceeded daily mean (WHO guidelines values), *N*, %	*N*/A	18, 0.50	688, 23.60	172, 5.29	*N*/A	*N*/A	*N*/A

Abbreviations: Atm. P., atmospheric pressure; IQR, interquartile range; *N*/A, not applicable; *N*/*D*, no data; NO_2_, nitrogen dioxide; PM2.5, particulate matter with a diameter of 2.5 μm or less; PM10 particulate matter with a diameter of 10 μm or less; RH, relative humidity; SD, standard deviation; SO_2_, sulfur dioxide; Temp., temperature; WHO, World Health Organization.

**Table 4 jcm-09-02351-t004:** Multivariable Poisson regression model. Effect of air pollution and weather conditions on mortality. Omnibus test, *p* < 0.001.

Variables		RR	Lower 95% CI for RR	Upper 95% CI for RR	*p*
Total mortality	NO_2_ µg/m^3^ * + Meteorological parameters	0.99	0.96	1.02	0.44
	SO_2_ µg/m^3^ * + Meteorological parameters	1.07	1.02	1.12	0.005
	PM2.5 µg/m^3^ ** + Meteorological parameters	1.01	0.99	1.01	0.40
	PM10 µg/m^3^ ** + Meteorological parameters	1.01	0.98	1.04	0.49
	Temp. °C ***	1.03	1.01	1.05	0.005
**Variables**		**RR**	**Lower 95% CI for RR**	**Upper 95% CI for RR**	***p***
Cardiovascular mortality	NO_2_ µg/m^3^ * + Meteorological parameters	1.01	0.97	1.05	0.78
	SO_2_ µg/m^3^ * + Meteorological parameters	1.02	0.99	1.05	0.12
	PM2.5 µg/m^3^ ** + Meteorological parameters	1.03	0.99	1.07	0.14
	PM10 µg/m^3^ ** + Meteorological parameters	0.99	0.95	1.03	0.58
	Temp. °C ***	1.05	1.02	1.09	<0.001

Abbreviations: CI, confidence interval; NO_2_, nitrogen dioxide; PM2.5, particulate matter with a diameter of 2.5 μm or less; PM10, particulate matter with a diameter of 10 μm or less; RR, relative risk; SO_2_, sulfur dioxide; Temp., temperature. * RR for 1-μg/m3 increase of gases; ** RR for 10-μg/m3 increase of particular matter; *** RR for 10 °C decrease of temperature.

**Table 5 jcm-09-02351-t005:** Multivariable Poisson regression model. Effect of air pollution and weather conditions on female mortality.

Variables		RR	Lower 95% CI for RR	Upper 95% CI for RR	*p*
Total mortality	NO_2_ µg/m^3^ * + Meteorological parameters	0.98	0.94	1.03	0.31
	SO_2_ µg/m^3^ * + Meteorological parameters	1.05	1.01	1.10	0.009
	PM2.5 µg/m^3^ ** + Meteorological parameters	0.99	0.96	1.03	0.73
	PM10 µg/m^3^ ** + Meteorological parameters	1.03	0.98	1.07	0.17
	Temp. °C ***	1.04	1.02	1.07	0.003
**Variables**		**RR**	**Lower 95% CI for RR**	**Upper 95% CI for RR**	***p***
Cardiovascular mortality	NO_2_ µg/m^3^ * + Meteorological parameters	0.99	0.93	1.05	0.69
	SO_2_ µg/m^3^ * + Meteorological parameters	1.02	0.99	1.05	0.19
	PM2.5 µg/m^3^ ** + Meteorological parameters	1.00	0.95	1.05	0.86
	PM10 µg/m^3^ ** + Meteorological parameters	1.03	0.97	1.09	0.36
	Temp. °C ***	1.08	1.04	1.13	<0.001

Abbreviations: CI, confidence interval; NO_2_, nitrogen dioxide; PM2.5, particulate matter with a diameter of 2.5 μm or less; PM10, particulate matter with a diameter of 10 μm or less; RR, relative risk; SO_2_, sulfur dioxide; Temp., temperature. * RR for 1-μg/m3 increase of gases; ** RR for 10-μg/m3 increase of particular matter; *** RR for 10 °C decrease of temperature.

**Table 6 jcm-09-02351-t006:** Multivariable Poisson regression model. Effect of air pollution and weather conditions on male mortality.

Variables		RR	Lower 95% CI for RR	Upper 95% CI for RR	*p*
Total mortality	NO_2_ µg/m^3^ * + Meteorological parameters	0.99	0.96	1.04	0.96
	SO_2_ µg/m^3^ * + Meteorological parameters	1.10	1.04	1.17	0.002
	PM2.5 µg/m^3^ ** + Meteorological parameters	1.03	0.99	1.06	0.13
	PM10 µg/m^3^ ** + Meteorological parameters	0.99	0.96	1.03	0.99
	Temp. °C ***	1.01	0.99	1.04	0.29
**Variables**		**RR**	**Lower 95% CI for RR**	**Upper 95% CI for RR**	***p***
Cardiovascular mortality	NO_2_ µg/m^3^ * + Meteorological parameters	1.02	0.97	1.07	0.40
	SO_2_ µg/m^3^ * + Meteorological parameters	1.02	1.00	1.04	0.08
	PM2.5 µg/m^3^ ** + Meteorological parameters	1.07	1.02	1.12	0.01
	PM10 µg/m^3^ ** + Meteorological parameters	0.95	0.90	1.01	0.60
	Temp. °C ***	1.02	0.98	1.07	0.25

Abbreviations: CI, confidence interval; NO_2_, nitrogen dioxide; PM2.5, particulate matter with a diameter of 2.5 μm or less; PM10, particulate matter with a diameter of 10 μm or less; RR, relative risk; SO_2_, sulfur dioxide; Temp., temperature. * RR for 1-μg/m3 increase of gases; ** RR for 10-μg/m3 increase of particular matter; *** RR for 10 °C decrease of temperature.

**Table 7 jcm-09-02351-t007:** Ratios of relative risks of effect of air pollution and weather conditions on mortality. Comparison of women and men.

Variables		RRR	Lower 95% CI for RRR	Upper 95% CI for RRR	*p*
Total mortality	NO_2_ µg/m^3^ + Meteorological parameters	0.99	0.93	1.05	0.74
	SO_2_ µg/m^3^ + Meteorological parameters	1.05	0.97	1.13	0.21
	PM2.5 µg/m^3^ + Meteorological parameters	1.04	0.99	1.09	0.11
	PM10 µg/m^3^ + Meteorological parameters	0.96	0.91	1.02	0.17
	Temp. °C	0.97	0.94	1.01	0.09
**Variables**		**RRR**	**Lower 95% CI for RRR**	**Upper 95% CI for RRR**	***p***
Cardiovascular mortality	NO_2_ µg/m^3^ + Meteorological parameters	1.03	0.95	1.11	0.45
	SO_2_ µg/m^3^ + Meteorological parameters	1.00	0.97	1.03	0.99
	PM2.5 µg/m^3^ + Meteorological parameters	1.07	1.00	1.14	0.53
	PM10 µg/m^3^ + Meteorological parameters	0.92	0.85	1.00	0.54
	Temp. °C	0.94	0.89	1.00	0.64

Abbreviations: CI, confidence interval; NO_2_, nitrogen dioxide; PM2.5, particulate matter with a diameter of 2.5 μm or less; PM10, particulate matter with a diameter of 10 μm or less; RRR, ratio of relative risk; SO_2_, sulfur dioxide; Temp., temperature.

## Data Availability

The data that support the findings of this study are available on request from the corresponding author, Ł.K.

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
