# Peer review of "Gender Differences in Association between Air Pollution and Daily Mortality in the Capital of the Green Lungs of Poland–Population-Based Study with 2,953,000 Person-Years of Follow-Up"

_jcm, 2020, doi:10.3390/jcm9082351_

Round 1
Reviewer 1 Report
This paper focuses on the association between SO2, NO2, PM2.5, PM10 and mortality, and on effect modifications due to gender, on a long span of time (2008-2017) in a middle-sized city of Poland, that is characterized by low air pollutants concentrations.
Even though I think that it is very interesting to explore how low levels of pollutants can affect mortality, and if the well-known differences in gender susceptibility change at such low levels, I think that this paper needs consistent editing in order to be considered for publication.
MAJOR REVISIONS
Abstract
I would suggest to add a synthetic description of the study design and of the statistical methods that were used.
Lines 21-24. This part could be eliminated.
Aim of the study
It is unclear whether this study focuses on short- or long-term effects of pollution. I would suggest to clarify it.
Patients and methods
- Lines 91-92. Authors state that they determined mean daily and yearly statistics of death, but it is unclear which outcome they used in the statistical analysis: did they use the daily number of deaths, or the yearly number of deaths? I would suggest to explain it.
- Lines 108. Authors state that they determined mean daily and yearly concentration of pollutants, but it is unclear what concentrations they used as predictors in the statistical models. I would suggest to clarify it.
- Lines 121-130. Unfortunately, I think that the model description needs consistent improvement. Authors should state if they are focusing on short-term effects or long-term effects, as Poisson regression model per se could be used for both purposes. Furthermore, authors stated that they used Poisson regression, leading the reader to think that they carried out a time-series analysis, but they then state that they use a time-stratified approach, that does not focus on time-series but rather on individual data. Do they refer to a conditional Poisson model, that has been proposed as a way of carrying out case-crossover studies?
If they are focusing on short-term exposure, did they evaluate the role of lagged exposures or just the exposure on the day of death? Why did they choose to include the change in maximum temperature from the previous day as a confounder? What did they adjust for in the models assessing the effects of temperature?
- Line 133. It is unclear what variation in temperature was used in the display of results. Was it 10°C or the difference between 5th and 95th percentile?
- In the statistical analysis paragraph authors should state that they carried out the analyses stratified by gender, and I would suggest to clarify why they decided to carry out separate analyses instead of adding an interaction term in the Poisson models. Indeed, such term would have allowed to test if differences between genders were statistically significant. The stratified analyses that is described in methods and results report the estimated RR in the two genders, but does not test for statistical differences. Since point estimates are very close to each other, it is hard to understand if the observed difference is significant or not.
Discussion
I would suggest to revise the discussion, as it seems a little unfocused.
- Lines 216-279 could be shortened consistently, as they refer to spatial differences in death rates and life expectancies that are beyond the scope of this article.
- Lines 286-289. Authors state that the influence of SO2 on mortality is higher among men, but given the large overlay of confidence intervals of the stratified estimates, I would suggest to revise this statement. Alternatively, authors could add a test to assess the difference between the two RRs.
A similar consideration applies to all stratified estimates: confidence intervals always overlay to some extent. Therefore it is difficult to highlight real differences in the effects.
Conclusions
Lines 361-362. I would suggest to revise this statement after testing the significance of the effect modification.
MINOR REVISIONS
Background
Line 46. I would suggest not to define air pollution as the main risk factor for public health, but as one of the main risk factors for public health.
Lines 54-60. I would suggest adding a citation.
Lines 61-62. I would suggest adding a citation.
Patients and methods
Lines 80-83. I think it would be nice to add a reference or a link for the cited project.
Lines 88-90. How many causes of deaths are listed in the death certificate? Did the authors use the main one for the analysis?
Results
Lines 151-167. I would suggest to shorten these paragraphs.
Table 1. I would suggest to report only the overall mean age, without stratifying by year. Yearly results might be moved to supplementary material.
Table 2. It is unclear to me if the cardiovascular mortality includes chronic ischemic heart disease, cerebral infarction, acute myocardial infarction and heart failure. If so, I would report them as subcategories of cardiovascular mortality.
Table 3. I would suggest to report only the overall data. Yearly results might be moved to supplementary material.
Table 4. I would suggest to move it to supplementary material.
Study limitations
Lines 349-351. Even though it is true that pollutant levels fluctuate within a city, monitoring stations should be representative of the average daily concentration of a city, if properly placed. What kind of monitoring stations were included in the study?
Conclusions
Lines 360-362. It is possible that no effect on cardiovascular mortality was observed because the sample of cases was not very big.
Author Response
Reviewer 1
This paper focuses on the association between SO2, NO2, PM2.5, PM10 and mortality, and on effect modifications due to gender, on a long span of time (2008-2017) in a middle-sized city of Poland, that is characterized by low air pollutants concentrations.
Even though I think that it is very interesting to explore how low levels of pollutants can affect mortality, and if the well-known differences in gender susceptibility change at such low levels, I think that this paper needs consistent editing in order to be considered for publication.
Dear Reviewer
Thank you very much for considering our manuscript for publication in JCM. We are grateful for the time and effort invested in reviewing our manuscript and for the many thoughtful comments and suggestions from the reviewers. We have considered these comments carefully and feel that we have fully addressed the reviewer’s concerns. We have replied to each comment below and edited the manuscript accordingly. We believe that this input has improved this manuscript greatly. Complying with all the suggestions, we hope that you will find the manuscript suitable for publication in JCM.
On behalf of the authors
Łukasz Kuźma MD
MAJOR REVISIONS
Abstract
I would suggest to add a synthetic description of the study design and of the statistical methods that were used.
Following sentence was added:
Time series regression with Poisson distributed model was used in statistical analysis.
Lines 21-24. This part could be eliminated.
Thank you for your comments. We have corrected the manuscript as recommended
Aim of the study
It is unclear whether this study focuses on short- or long-term effects of pollution. I would suggest to clarify it.
We thank the Reviewer for this comment. We rewrote the aim of the study. Currently it is:
PURPOSE This study was carried out to assess the short-term effect of the atmospheric conditions and air pollutants on mortality in Białystok, the capital of Green Lungs of Poland
Patients and methods
- Lines 91-92. Authors state that they determined mean daily and yearly statistics of death, but it is unclear which outcome they used in the statistical analysis: did they use the daily number of deaths, or the yearly number of deaths? I would suggest to explain it.
Appropriate clarification was provided in the manuscript.
Time series regression was used to assess the short-term effect of particles and gases concentration and temperature on mortality. We use multivariable Poisson distributed model. To minimize the collinearity effect, each air pollutant was modelled individually. Separate models were created for temperature with humidity and atmospheric pressure as covariates. The daily number of deaths was defined as dependent variable.
We have added this information into the methods section, under “Material and method” section.
- Lines 108. Authors state that they determined mean daily and yearly concentration of pollutants, but it is unclear what concentrations they used as predictors in the statistical models. I would suggest to clarify it.
Appropriate clarification was provided in the manuscript. See Materia and Method section.
- Lines 121-130. Unfortunately, I think that the model description needs consistent improvement. Authors should state if they are focusing on short-term effects or long-term effects, as Poisson regression model per se could be used for both purposes. Furthermore, authors stated that they used Poisson regression, leading the reader to think that they carried out a time-series analysis, but they then state that they use a time-stratified approach, that does not focus on time-series but rather on individual data. Do they refer to a conditional Poisson model, that has been proposed as a way of carrying out case-crossover studies?
Time series regression was used to assess the short-term effect of particles and gases concentration and temperature on mortality. We use multivariable Poisson distributed model. To minimize the collinearity effect, each air pollutant was modelled individually. Separate models were created for temperature with humidity and atmospheric pressure as covariates. The daily number of deaths was defined as dependent variable. The independent variables were daily concentrations of NO2, SO2, PM2.5, PM10 and temperature.
According to Krishnan Bhaskaran et al. publication “Time series regression studies in environmental epidemiology” in “International Journal of Epidemiology” to control for the long-term trend and seasonal effects we use time stratified model (simple indicator variables method). To avoid long term trends, we use elapsed calendar month as the time interval for daily data which resulted in creating 120 strata.
If they are focusing on short-term exposure, did they evaluate the role of lagged exposures or just the exposure on the day of death? Why did they choose to include the change in maximum temperature from the previous day as a confounder?
We focused only on the day of exposure and didn’t evaluate the role of lagged exposures.
We use the day to day changes in temperature, because it has been identified as environmental risk factor for acute coronary syndromes and mortality in previous studies.
Lashari et al., Yamaji et al., Wolf et al pointed out, that sudden temperature change may be as important as low temperature, in terms of having an effect on occurrence of ACS. Similar conclusions were drawn in a study that examined seasonal variation in the occurrence of acute aortic dissection, which has comparable pathophysiological mechanisms to ACS triggered by rapidly changes in weather (Mehta et al.). Data from recent study conducted in Northeast China also showed seasonal variation confirming the effect of low air temperature having a big impact on admissions during warm season.
Bibliogrpahy:
Lashari et al. (Variation in admission rates of acute coronary syndrome patients in coronary care unit accoding to different seasons, 2015)
Yamaji et al. (Relation of ST-Segment Elevation Myocardial Infarction to Daily Ambient Temperature and Air Pollutant Levels in a Japanese, 2017)
Wolf et al. (Air temperature and the occurrence of myocardial infarction in Augsburg, Germany. Circulation. 2009)
Mehta et al. The winter peak in the occurrence of acute aortic dissection is independent of climate
Xue, Association Between Air Temperature and the Incidence of Acute Coronary Heart Disease in Northeast China
What did they adjust for in the models assessing the effects of temperature?
In temperature models we use the humidity and atmospheric pressure as covariates. As in the others models to control for the long-term trend and seasonal effects we use time stratified model (simple indicator variables). We also adjusted temperature model for day of the week and public holidays.
- Line 133. It is unclear what variation in temperature was used in the display of results. Was it 10°C or the difference between 5thand 95th percentile?
The results were presented as risk ratios for 10°C decrease in temperature and CI from the 5th to the 95th percentile. We have rewritten this information in the methods section, under “Material and method” section
- In the statistical analysis paragraph authors should state that they carried out the analyses stratified by gender, and I would suggest to clarify why they decided to carry out separate analyses instead of adding an interaction term in the Poisson models. Indeed, such term would have allowed to test if differences between genders were statistically significant. The stratified analyses that is described in methods and results report the estimated RR in the two genders, but does not test for statistical differences. Since point estimates are very close to each other, it is hard to understand if the observed difference is significant or not.
- Due to differences in death rates and other causes of death, we decided to create separate models for specific sexes to better show gender differences. In the first stage, we created a model for all patients. In the second stage, separate models for male and females. In third stage we compare head to head impact of air pollution on women and men. The differences in the risk ratio between males and females were calculated according to Altman method [16]. Results were presented as ratio of relative risk (RRR) and CI from the 5th to the 95th percentile.
We have rewritten this information in the methods section, under “Material and method” section and added new information in “Results” and “Discussion” section.
Discussion
I would suggest to revise the discussion, as it seems a little unfocused.
- Lines 216-279 could be shortened consistently, as they refer to spatial differences in death rates and life expectancies that are beyond the scope of this article.
Thank you for your attention. We have removed information that is not relevant to the discussion
Lines 286-289. Authors state that the influence of SO2 on mortality is higher among men, but given the large overlay of confidence intervals of the stratified estimates, I would suggest to revise this statement. Alternatively, authors could add a test to assess the difference between the two RRs. A similar consideration applies to all stratified estimates: confidence intervals always overlay to some extent. Therefore it is difficult to highlight real differences in the effects.
- We've added analysis results for the entire population and a gender comparison [Table 4] and [Table 7]. SO2 concentration was found to have an influence on total mortality in the whole population (RR 1.07, 95% CI 1.02–1.12; P=0.005). This effect was more pronounced in the male population; the RR for 1-mg/m3 increase of SO2 was 1.10 (95% CI 1.04–1.17; P=0.002) but the difference between males and females wasn’t significant RRR 1.05 (95% CI 0.97 –1.13; P=0.21)
We've added analysis of the gender comparison [Table 7]. Additionally, we’ve included the results in discussions and conclusions sections.
Conclusions
Lines 361-362. I would suggest to revise this statement after testing the significance of the effect modification.
We have rewritten the Conclusion section according to the new results in Table 7.
MINOR REVISIONS
Background
Line 46. I would suggest not to define air pollution as the main risk factor for public health, but as one of the main risk factors for public health.
Thank you for your comment. We made appropriate adjustments
Lines 54-60. I would suggest adding a citation.
Thank you for your comment. We have added citation. References No 10
Lines 61-62. I would suggest adding a citation.
Thank you for your comment. We have added citation. References No 11 and 12.
Lines 80-83. I think it would be nice to add a reference or a link for the cited project.
We have add the references We have added citation. References No 14.
Lines 88-90. How many causes of deaths are listed in the death certificate? Did the authors use the main one for the analysis?
In Poland, the physician certifying death is required to describe and provide direct, secondary and primary cause of death. Then, Death Cards prepared in this way are verified at the Central Statistical Office in Olsztyn by coders. Based on the description, these specialists code the cause of death. It should be emphasized that the ICD-10 code of the initial cause of death is entered by the coder physician in the statistical set and this data was used in the statistical analysis.
Lines 151-167. I would suggest to shorten these paragraphs.
As recommended, we have shortened the paragraphs
Table 1. I would suggest to report only the overall mean age, without stratifying by year. Yearly results might be moved to supplementary material.
As recommended, we have shortened the table and created the Table 9 and moved this to supplementary material chapter.
Table 2. It is unclear to me if the cardiovascular mortality includes chronic ischemic heart disease, cerebral infarction, acute myocardial infarction and heart failure. If so, I would report them as subcategories of cardiovascular mortality.
As recommended, we've reformatted the Table 2
Table 3. I would suggest to report only the overall data. Yearly results might be moved to supplementary material.
As recommended, we have shortened the table and created the Table 10 and move this to supplementary material chapter.
Table 4. I would suggest to move it to supplementary material.
As recommended, we moved the table to supplementary material chapter
Study limitations
Lines 349-351. Even though it is true that pollutant levels fluctuate within a city, monitoring stations should be representative of the average daily concentration of a city, if properly placed. What kind of monitoring stations were included in the study?
In Białystok the data on the concentrations of NO2 SO2 (except 2013) and PM2.5 for 2009 and 2015–2017 were obtained for Waszyngtona Street 16 station (ID: PL0148A, GPS coordinates: 53°12′ N, 23°15 ′ E). The SO2 concentration for 2013 was obtained from PodMiejska station (ID: PL0149A, GPS: 53°13′ N, 23°22′ E) and PM2.5 for 2010-2014 was obtained from Warszawska Street 75A station (ID: PL0496A, GPS: 53°12′ N, 23°18′ E).The data on the concentrations of PM10 were obtained for 2009 – 2014 and for 2015 – 2017. In case in PM10 the date was obtained from Legionowa 8 station for 2008 and 2009 (ID: PL0147A, GPS: 53°13′ N, 23°15′ E), Waszyngtona Street 16 station for 2010 and for 2011 – 2017 from Warszawska Street 75A station. All station except PodMiejska are located in the city center and raccording to Voivodeship Inspectorate for Environmental Protection in Białystok menasurment there are representative of the average daily concentration of a city,
However, in our opinion, in particular in areas with family buildings, where many houses are heated with solid fuel, new stations should be created whose value could be different due to the local nature of the district
Conclusions
Lines 360-362. It is possible that no effect on cardiovascular mortality was observed because the sample of cases was not very big.
Thank you for your comment. We rewrote the conclusions according to your comments.
Reviewer 2 Report
The aim of the paper was to evaluate the health impact for the years 2008-2017 from some air pollutants (SO2, NO2, PM10, PM2.5) and atmospheric conditions (temperature, relative humidity and atmospheric pressure) on male and female residents of the city of Bialystok. Bialystok is defined as the capital of the Green Lungs of Poland, an area characterized by very low air pollution levels and a clean environment. The health impact from air pollutants and atmospheric conditions was measured in terms of daily and annual total and cardiovascular mortality and statistical associations were investigated. Moreover, comparison between male and female proportions for the main specific causes of death were carried out for the overall period.
The strength points of the paper are: 1) the assessment of health impact, in terms of total and cardiovascular mortality, from both air pollution and climatic parameters; 2) the investigation carried out on a city with relatively low air pollution, differently from the majority of the existing studies on such topic.
Areas of weakness include: 1) comparisons between male and female temporal mortality levels and age trends were carried out on the number of deaths without considering the dimension of the populations who originated such numbers; indeed mortality levels are usually expressed in terms of standardized mortality rates and not of absolute numbers; 2) some confusion comes out between the main objective, which is the association of mortality with air pollution and atmospheric conditions, and the very long and detailed description of the main differences in mortality detected between males and females; 3) the lack of inclusion of other important pollutants such as O3 and IPA that could have also been responsible for the observed mortality levels; 4) the “Discussion” is not well organised and rather dispersive.
I suggest to substitute the term “norm” related to the air pollutant WHO reference values with “WHO guidelines values” as correctly mentioned in line 109 or “WHO reference values”.
Introduction:
The “Introduction” (Background and Aim of the Study) is rather poor in terms of “state of the art” and aim of the study. In my opinion, the two main objectives of such study (associations with pollutants and comparison between males and females mortality) should be better declared. Moreover, several paragraphs should move here from the “Discussion” as detailed below.
Patients and methods:
Did you include all air pollutants monitored from the meteorological stations? Some other pollutants, if available, could be included.
Line 77: it would be interesting to know what industrial plants are located in the city.
Results:
All tables are not self-explanatory.
In order to facilitate the reader, I suggest to cite each Table in the text at the beginning of the presentation of the relative data.
I also suggest to add the X ICD code you selected for the different causes of death in the first Table or in “Methods”.
Line 139-147: You said that the number and the age of people who died increased during the investigated period in both male and female populations. It seems that the dimension and the age-classes distributions of the resident populations, from whom such deaths generated, were not considered. In order to compare mortality among different period or different locations, standardised mortality rates are usually used. Moreover, if both sexes showed the same increasing tendency, why not simplify in the text with only one sentence?
Line 150: data reported here refer to both Table 1 and Table 2.
Table 1 and Table 2 : more explanations should be given in the legends or in the text. In my opinion, it is even not clear what the p values refer to, but I suppose to the statistical difference between sexes.
Table 2: what do you mean with “Senility”?
Line 168-169: the strong positive correlation between PM2.5 and PM10 is rather obvious because PM2.5 is a subset of PM10.
Table 4: I suggest to add in the title that the correlations are “between physical variables and air pollutants”.
Line 177: I suggest to add “an increase of 1mg/m3 of SO2”.
Line 184: the RR is for 10mg/m3 and not for 1mg/m3.
Discussion:
Most “Discussion” is focused on collateral topics (aging, life expectancy, differences in the percentages of causes of death between Countries) and not on the main objective of the study which is, as stated in the “Abstract” and in the “Aim of the study”, “to assess the influence of the atmospheric conditions and air pollution on mortality in Bialystok”. In my opinion the real “Discussion” starts from line 280 and all the previous paragraphs should be shortened.
I suggest to avoid repetition of data already shown in “Results”, if not necessary.
You introduced the Standardised Death Rates (SDRs) in the “Discussion”, giving some numerical data . I suggest to move such data in the “Results” and compare them with the EU and Poland data in the “Discussion”.
Line 213-215: I suggest to move this sentence to “Methods 3.1”
215-220: I suggest to move to the “Introduction”.
Line 224: I suggest to substitute “were reflected” with “are aligned with”.
Line 235-237: the number of deaths was already given in the “Results”. I suggest to move the death rates reported here to the “Results”. Moreover, it is not clear whether they are standardized death rates or not. If this is not the case, comparisons with the other reported values (which are standardized) could be misleading and, in my opinion, this limitation should be at least mentioned.
Line 286: which “thesis” is confirmed by the results?
Line 290-291: the amount of SO2 increase should be mentioned.
Line 297-298: in the present study the “association” between SO2 and lung cancer was not assessed.
Line 299-300: if you decide to repeat all the specific data (but it seems unnecessary), you should also mention the 1mg/m3 increased level of SO2
Line 315: which “plaque” do you mean?
Line 322-325: not clear enough
Line 326: please clarify “are not unambiguous”
Conclusion:
Conclusion is rather poor. It must be improved.
Line 357-358: on what do you base such an affirmation? Where mortality rates really used to assess the associations?
Line 365-368: the WHO values are “guidelines” and not “safe range”.
Author Response
Reviewer 2
The aim of the paper was to evaluate the health impact for the years 2008-2017 from some air pollutants (SO2, NO2, PM10, PM2.5) and atmospheric conditions (temperature, relative humidity and atmospheric pressure) on male and female residents of the city of Bialystok. Bialystok is defined as the capital of the Green Lungs of Poland, an area characterized by very low air pollution levels and a clean environment. The health impact from air pollutants and atmospheric conditions was measured in terms of daily and annual total and cardiovascular mortality and statistical associations were investigated. Moreover, comparison between male and female proportions for the main specific causes of death were carried out for the overall period.
The strength points of the paper are: 1) the assessment of health impact, in terms of total and cardiovascular mortality, from both air pollution and climatic parameters; 2) the investigation carried out on a city with relatively low air pollution, differently from the majority of the existing studies on such topic.
Areas of weakness include: 1) comparisons between male and female temporal mortality levels and age trends were carried out on the number of deaths without considering the dimension of the populations who originated such numbers; indeed mortality levels are usually expressed in terms of standardized mortality rates and not of absolute numbers; 2) some confusion comes out between the main objective, which is the association of mortality with air pollution and atmospheric conditions, and the very long and detailed description of the main differences in mortality detected between males and females; 3) the lack of inclusion of other important pollutants such as O3 and IPA that could have also been responsible for the observed mortality levels; 4) the “Discussion” is not well organised and rather dispersive.
I suggest to substitute the term “norm” related to the air pollutant WHO reference values with “WHO guidelines values” as correctly mentioned in line 109 or “WHO reference values”.
Dear Sir or Madame!
Thank you very much for considering our manuscript for publication in JCM. We are grateful for the time and effort invested in reviewing our manuscript and for the many thoughtful comments and suggestions from the reviewers. We have considered these comments carefully and feel that we have fully addressed the reviewer’s concerns. We have replied to each comment below and edited the manuscript accordingly. We believe that this input has improved this manuscript greatly. Complying with all the suggestions, we hope that you will find the manuscript suitable for publication in JCM.
On behalf of the authors
Łukasz Kuźma MD
Introduction:
The “Introduction” (Background and Aim of the Study) is rather poor in terms of “state of the art” and aim of the study. In my opinion, the two main objectives of such study (associations with pollutants and comparison between males and females mortality) should be better declared. Moreover, several paragraphs should move here from the “Discussion” as detailed below
We are very grateful for the time and effort invested in reviewing our manuscript..We have made the changes in accordance with the recommendations, detailed changes have been listed below in the discussion section, summarizing the individual comments. We've improved the introduction chapter.
Patients and methods:
Did you include all air pollutants monitored from the meteorological stations? Some other pollutants, if available, could be included.
We used all available continuous measurements in the years 2008 - 2017. In these years Pb and C6H6 measurements were also carried out, but these measurements were carried out periodically for several days a year.
Line 77: it would be interesting to know what industrial plants are located in the city.
We thank the reviewer for this question. The main sources of pollution in analyzed years were Power Plant Białystok S.A., Municipal Thermal Energy Company Sp. z o.o. w Białymstoku, and factories „BISONOL-BIAL” S.A. and Biaglass Glassworks Białystok Sp. z o.o. and BIAFORM S.A.,At present, individual industrial plants are located in the city (the biggest plywood manufacture and glassworks factory)
Results:
All tables are not self-explanatory.
In order to facilitate the reader, I suggest to cite each Table in the text at the beginning of the presentation of the relative data. I also suggest to add the X ICD code you selected for the different causes of death in the first Table or in “Methods”.
We thank the reviewer for this comment. We have rewritten the Results section
In Material and Method section we’ve added this information:
We used the ICD-10 codes to define deaths due to CVDs (I00–I99), including the major ones such as ischemic heart disease (I2X), stroke (I6X), and heart failure (I5X)
Line 139-147: You said that the number and the age of people who died increased during the investigated period in both male and female populations. It seems that the dimension and the age-classes distributions of the resident populations, from whom such deaths generated, were not considered. In order to compare mortality among different period or different locations, standardised mortality rates are usually used. Moreover, if both sexes showed the same increasing tendency, why not simplify in the text with only one sentence?
Thank you for your very relevant remarks. These data with the recommendations of the first reviewer, were transferred to additional materials. We have removed unnecessary information from Results chapter.
Line 150: data reported here refer to both Table 1 and Table 2.
We apologize for missing this and thank the reviewer for this suggestion. We fixed this.
Table 1 and Table 2 : more explanations should be given in the legends or in the text. In my opinion, it is even not clear what the p values refer to, but I suppose to the statistical difference between sexes.
Thank you for your attention. Yes., in Table 1 and 2 the values relate to differences in the percentage of deaths between the sexes. Regarding the reviewer's comments, the Table 1 and 2 has been remodeled. We added the information about p – values.
Table 2: what do you mean with “Senility”?
The therm senility is related to R54 ICD code. According to ICD10 coding rules R54 code is applicable to old age, senile asthenia, senile debility, frailty and senescence
Line 168-169: the strong positive correlation between PM2.5 and PM10 is rather obvious because PM2.5 is a subset of PM10.
We apologize for missing this. We have removed this information. Additionaly Table 4 was moved to Supplementary Material.
Table 4: I suggest to add in the title that the correlations are “between physical variables and air pollutants”.
Thank you for your attention, we have revised this information. The Table 4 according to reviewer 1 was moved to supplementary material.
Line 177: I suggest to add “an increase of 1mg/m3 of SO2”.
We apologize for missing this. We have corrected the manuscript as recommended
Line 184: the RR is for 10mg/m3 and not for 1mg/m3.
Thank you for your comments. We have corrected the manuscript as recommended
Discussion:
Most “Discussion” is focused on collateral topics (aging, life expectancy, differences in the percentages of causes of death between Countries) and not on the main objective of the study which is, as stated in the “Abstract” and in the “Aim of the study”, “to assess the influence of the atmospheric conditions and air pollution on mortality in Bialystok”. In my opinion the real “Discussion” starts from line 280 and all the previous paragraphs should be shortened.
Thank you for your opinion, we agree and shortened the first paragraphs of discussion
I suggest to avoid repetition of data already shown in “Results”, if not necessary.
We have rewritten the discussions section and removed unnecessary results
You introduced the Standardised Death Rates (SDRs) in the “Discussion”, giving some numerical data . I suggest to move such data in the “Results” and compare them with the EU and Poland data in the “Discussion”.
We’ve rewritten the discussions section and removed moved some paragraphs to results section
Line 213-215: I suggest to move this sentence to “Methods 3.1”
We moved this sentence to Methods 3.1 section.
215-220: I suggest to move to the “Introduction”.
We have shortened the paragraphs and moved sentences to the introduction
Line 224: I suggest to substitute “were reflected” with “are aligned with”.
We have made this correction
Line 235-237: the number of deaths was already given in the “Results”. I suggest to move the death rates reported here to the “Results”. Moreover, it is not clear whether they are standardized death rates or not. If this is not the case, comparisons with the other reported values (which are standardized) could be misleading and, in my opinion, this limitation should be at least mentioned.
We clearly demarcated in the discussion section, when we used standardized death rate and when we used plain death rate. We have highlighted this in the limitations section
Line 286: which “thesis” is confirmed by the results?
We corrected this sentence. The results obtained in the present study seem to confirm previously study
Line 290-291: the amount of SO2 increase should be mentioned.
Correction has been made.
Line 297-298: in the present study the “association” between SO2 and lung cancer was not assessed. Line 299-300: if you decide to repeat all the specific data (but it seems unnecessary), you should also mention the 1mg/m3 increased level of SO2
We’ve rewritten and correct mistake in this section according to recommendation We also add 1 mg/m3 whenapplicable.
Line 315: which “plaque” do you mean?
We’ve corrected the term to “atherosclerotic plaque”.
Line 322-325: not clear enough
We’ve tried to clarify this section: “Despite the higher percentage of deaths due to CVDs in the female population, no influence of PM2.5 on deaths was observed in this group. This can be explained by the lower contribution of sudden cardiac deaths observed in the female population (4.1% vs. 10.3% of deaths due to acute coronary event and sudden deaths).”
Line 326: please clarify “are not unambiguous”
The unclarity of this part is a result of poor word choice during translation. After revision and additional calculation we have changed the section to “Some previous studies have demonstrated that the female population is more susceptible to the influence of particulate matter [39,40]; however, the literature is dominated by reports showing a greater impact of air pollution on male population or the lack of differences between the sexes [41–44] and results of our study fall in this last trend.
Conclusion:Conclusion is rather poor. It must be improved.
We have changed the conclusion section including points 1 to 3.
Line 357-358: on what do you base such an affirmation? Where mortality rates really used to assess the associations?
Our conclusion is based on the results presented in tables 4 to 6, where we can find statistically significant correlation between SO2, PM2,5, temperature and mortality. After reanalysing our results and additional calculations, we have removed the part “and this effect was more pronounced in the male population.”
Line 365-368: the WHO values are “guidelines” and not “safe range”.
We have corrected the according conclusion.
Round 2
Reviewer 1 Report
I think that the authors properly addressed my concerns.
I appreciated that they introduced a statistical comparison between the effects estimated among males and females: conclusions are now supported by data.
Author Response
We are thankful for the opportunity to publish our work.
On behalf of all Authors
Łukasz Kuźma
Reviewer 2 Report
The Authors followed most of the punctual suggestions and the manuscript has improved. However, the suggested shift of some paragraphs to different parts of the manuscript did not always consider the context in which they were added. So there are some repetitions and some paragraphs are not well organized.
As far as the “Background” and “Aim of the Study” are concerned, they were both untouched, and thereafter they remained rather poor.
Also the paragraphs in the “Discussion” on aging, life expectancy and differences in the percentages of causes of death between countries remained untouched and, in my opinion, they are too long considering the primary aim of the study.
The suggestion of substituting the term “norm” related to the WHO values for air pollutants with the more correct “WHO guidelines values” was not taken into account.
Table 9 of Supplementary material is never mentioned in the text.
Abstract: some grammatical and punctuation mistakes are present in the Conclusions of the Abstract.
Line 30: the affirmation that there were no gender differences in the effect of air pollution is correct if referred to SO2 and total mortality. But if you also mention PM2.5 (even though with punctuation mistakes) a significant effect on cardiovascular mortality was detected only in males.
Introduction: the “Background” and “Aim of the Study” were both untouched and so they remained rather poor.
3.2 Mortality data: a reorganization of this paragraph seems necessary. The population number, already partially reported in 3.1, should move there completely. Repetitions related to the ICD 10 causes of death classification should be avoided (line 85-87 and line 90-92).
Results:
Line 142-148: there are several repetitions, e.g. the number of deaths in males and females.
Line 145: not “a rate of deaths” but “the rate of deaths” or even better “the death-rate”.
Line 166: Table 10 of Supplementary material reports annual means and is not aligned to daily values reported in line 160-165.
Line 167: “between that”?
Line 174: “have shown”? “show”?
Author Response
The Authors followed most of the punctual suggestions and the manuscript has improved. However, the suggested shift of some paragraphs to different parts of the manuscript did not always consider the context in which they were added. So there are some repetitions and some paragraphs are not well organized.
Thank you very much for considering our manuscript for publication. Complying with all the suggestions, we hope that you will find the manuscript suitable for publication in JCM. The Background and Aim of the Study has been rephrased. The Discussion has been shortened. Detailed corrections regarding your minor comments have been made. In line with your recommendation we asked for native speaker support, what, we believe, has improved the flow of the text and its readability.
As far as the “Background” and “Aim of the Study” are concerned, they were both untouched, and thereafter they remained rather poor.
Thank you for bringing our attention to those sections. We have rephrased the aim of the study and reorganized the background section. We have joined the background section with chapter 3.1 to avoid repetition in different sections of the article.
Also the paragraphs in the “Discussion” on aging, life expectancy and differences in the percentages of causes of death between countries remained untouched and, in my opinion, they are too long considering the primary aim of the study.
Thank you for your comments. We have shortened the discussion section according to your suggestions, but we have left some data to show the Białystok data in comparison with the population of Poland.
The suggestion of substituting the term “norm” related to the WHO values for air pollutants with the more correct “WHO guidelines values” was not taken into account.
We are sorry for this omission. We have rephrased according paragraphs to avoid using of the „norm” term. One of the corrected sentences: „The daily upper limit for PM10 (50 µg/m3) recommended by the WHO guidelines was observed to be exceeded on 172 days (5.29%) [Table 3].”
We hope that this form is more acceptable.
Table 9 of Supplementary material is never mentioned in the text.
Thank you for bringing it to our attention. A reference to this table has been added in the Results section
In the case of women, an increase in the number of deaths was observed every year (1478 in 2008 vs. 1846 in 2017), and a clear trend of increase in the mean age of the deceased was also noted 74.10 (SD=16.67) in 2008 vs. 78.08 (SD=14.40) in 2017. Similar observations of the increased number of deaths and the increase in the age of the deceased were noted in the male population (1683, mean age 67.33 (SD=16.83) in 2008 vs. 1942, mean age 69.59 (SD=16.29) in 2017) [Table 9 in Supplementary material].
Abstract: some grammatical and punctuation mistakes are present in the Conclusions of the Abstract.
We have tried to find and correct all the mistakes. We hope that we succeeded in this task.
Line 30: the affirmation that there were no gender differences in the effect of air pollution is correct if referred to SO2 and total mortality. But if you also mention PM2.5 (even though with punctuation mistakes) a significant effect on cardiovascular mortality was detected only in males.
Thank you for highlighting this mistake. We have changed according part of the abstract and conclusions sections:
Air quality and atmospheric conditions had an impact on the mortality of Bialystok residents. The main air pollutant that influenced the mortality rate was SO2, and there were no gender differences in the impact of this pollutant. In the male population, an increased exposure to PM2.5 concentration was associated with significantly higher cardiovascular mortality. These findings suggest that improving air quality, in particular, even with lower SO2 levels than currently allowed by WHO guidelines, may benefit public health. Further studies on this topic are needed, but our results bring questions whether the recommendations concerning acceptable concentrations of air pollutants should be stricter, or is there a safe concentration of SO2 in the air at all.
Introduction: the “Background” and “Aim of the Study” were both untouched and so they remained rather poor.
Thank you for your comment. Did you mean the introduction section in abstract? We have changed the introduction and purpose sections of the abstract:
Abstract: INTRODUCTION: Air pollution is considered to be one of the main risk factors for public health.According to the European Environment Agency (EEA), air pollution contributes to the premature death of approximately 500,000 citizens of the European Union (EU), including almost 5,000 inhabitants of Poland, every year. PURPOSE To assess the gender differences in the impact of air pollution on the mortality in the population of the city of Bialystok - the capital of the Green Lungs of Poland.
3.2 Mortality data: a reorganization of this paragraph seems necessary. The population number, already partially reported in 3.1,should move there completely.
We have moved the data from the 3.1 chapter to other sections of the article – partially to the introduction section and partially to the Mortality Data.
See line 83-89
Repetitions related to the ICD 10 cause of death classification should be avoided (line 85-87 and line 90-92).
We are grateful for your opinion. Repetitions have been deleted.
Line 142-148: there are several repetitions, e.g. the number of deaths in males and females.
We are grateful for your opinion. Repetitions have been deleted.
Line 145: not “a rate of deaths” but “the rate of deaths” or even better “the death-rate”.
Thank you for your opinion. Needed changes have been made in whole manuscript.
Line 166: Table 10 of Supplementary material reports annual means and is not aligned to daily values reported in line 160-165.
We have added annual WHO guidelines values. We have made appropriate changes in the Methods and Results sections.
“The annual WHO guidelines values for PM2.5 and PM10 were exceeded every year [Table 10 in Supplementary material]”.
See line 168
Line 167: “between that”?
Thank you for your opinion. Needed changes have been made.
A moderate positive correlation was found between concentration of particulate matter and gases—PM2.5 vs. NO2:
See line 170
Line 174: “have shown”? “show”?
Thank you for your opinion. Needed changes have been made.
In Table 4 the impact of air pollution on short-term mortality in studied population is shown. A 1 mg/m3 increase of SO2 concentration (RR 1.07, 95% CI 1.02–1.12; P=0.005) and a 10°C decrease of temperature (RR 1.03, 95% CI 1.01–1.05; P=0.005) were related to an increase in the number of daily deaths.
See line: 177